# The Single-Stranded DNA-Binding Gene *Whirly* (*Why1*) with a Strong Pathogen-Induced Promoter from *Vitis pseudoreticulata* Enhances Resistance to *Phytophthora capsici*

**DOI:** 10.3390/ijms23148052

**Published:** 2022-07-21

**Authors:** Chengchun Lai, Qiuxia Que, Ruo Pan, Qi Wang, Huiying Gao, Xuefang Guan, Jianmei Che, Gongti Lai

**Affiliations:** 1Institute of Agricultural Engineering Technology, Fujian Academy of Agricultural Sciences, Fuzhou 350003, China; lccisland@163.com (C.L.); queqiuxia2020@163.com (Q.Q.); rrpanpan@163.com (R.P.); nkywq@163.com (Q.W.); faas14061407@163.com (H.G.); guan-619@163.com (X.G.); 2Institute of Agricultural Bio-Resources Research, Fujian Academy of Agricultural Sciences, Fuzhou 350003, China

**Keywords:** *Whirly*, promoter, *Vitis pseudoreticulata*, *Phytophthora capsici*, *PR* gene

## Abstract

*Vitis vinifera* plants are disease-susceptible while *Vitis pseudoreticulata* plants are disease-resistant; however, the molecular mechanism remains unclear. In this study, the single-stranded DNA- and RNA-binding protein gene *Whirly* (*VvWhy1* and *VpWhy1*) were cloned from *V. vinifera* “Cabernet Sauvignon” and *V. pseudoreticulata* “HD1”. *VvWhy1* and *VpWhy1* promoter sequences (*pVv* and *pVp*) were also isolated; however, the identity of the promoter sequences was far lower than that between the *Why1* coding sequences (CDSs). Both *Why1* gene sequences had seven exons and six introns, and they had a C-terminal Whirly conserved domain and N-terminal chloroplast transit peptide, which was then verified to be chloroplast localization. Transcriptional expression showed that *VpWhy1* was strongly induced by *Plasmopara viticola*, while *VvWhy1* showed a low expression level. Further, the GUS activity indicated *pVp* had high activity involved in response to *Phytophthora capsici* infection. In addition, *Nicotiana benthamiana* transiently expressing *pVp::VvWhy1* and *pVp::VpWhy1* enhanced the *P. capsici* resistance. Moreover, *Why1*, *PR1* and *PR10* were upregulated in *pVp* transgenic *N. benthamiana* leaves. This research presented a novel insight into disease resistance mechanism that *pVp* promoted the transcription of *Why1*, which subsequently regulated the expression of *PR1* and *PR10*, further enhancing the resistance to *P. capsici*.

## 1. Introduction

Grapevine (*Vitis* spp.) is an important fruit and beverage crop with extensive adaptability, and it is widely distributed and cultivated all over the world. *Vitis* plants are assigned to species originated in three regions, Eurasian, East Asian and North American [1], among which, *Vitis vinifera* is the only Eurasian species. However, table and wine grape varieties are mainly limited to *V. vinifera* conferring a high yield, good fruit quality and high economic value [2]. However, *V. vinifera* is confronted with major disease threats such as downy mildew, powdery mildew and anthracnose [3,4,5,6]. Downy mildew is one of the most serious diseases in grapevine [7,8], which is caused by *Plasmopara viticola*, and it could lead to the abuse of pesticides and fertilizers and bring great damage to the grape industry, consumers and the environment [9]. On the contrary, *Vitis pseudoreticulata* belonging to the East Asian species is a wild germplasm involved in the resistance to downy mildew [10]. So far, more than 31 *Rpv* (resistance to *P. viticola*) loci were identified from East Asian and North American grapevine species [11,12,13,14]. Moreover, an increasing number of disease resistance (or related) genes were identified and reported in several wild species including *V. pseudoreticulata*. A transcriptomic analysis of *V. pseudoreticulata* laid a foundation for further analysis of key genes involved in the resistance to downy mildew [15]. The overexpression of *VpSTS*29/*STS*2 in *V. vinifera* and *Arabidopsis* revealed that *VpSTS*29/*STS*2 enhanced fungal tolerance through a positive feedback loop [16]. *VpRPW*8 from *V. pseudoreticulata* was induced by *P. viticola*, and transgenic *Nicotiana benthamiana* improved the resistance to *Phytophthora capsici* [17,18]. Therefore, there is an immediate need for disease-resistance-associated gene discovery in *V. pseudoreticulata.*

Transcription factors play important roles in plant disease resistance via regulating the expression of resistance genes (*R* gene) or other constitutive genes. Whirly (Why) is a small family as single-stranded DNA-binding transcription factors in plants [19], and recent studies have shown that Why also had RNA-binding function [20]. Why proteins are dual-located on both the organelle (chloroplast/mitochondria) and the nucleus with DNA- and RNA-binding features [21,22]; among these proteins, Why1 has versatile functions involved in pathogen-induced transcription [14], embryonic development [23], abiotic stress response [24,25,26], genome repair [27], telomere maintenance [28] and leaf senescence [29]. Potato PBF-2 (*PR10a* binding factor 2), renamed as StWhy1 subsequently, was the first identified Why family member in plants [19]. StWhy1 has been implicated in the *PR-10a* activation binding to the ERE (elicitor response element) [19,30], and is required for SA-dependent disease resistance. Pathogenesis-related (PR) proteins were first reported in tobacco-mosaic-virus-infected *Nicotiana tabacum* plants [31]. Pathogenesis-related genes encoding PR proteins were induced by pathogen or abiotic stimulation in host plants [32]. The *PR* gene can be regulated by Why1 involved in disease resistance, but its function differentiation remains unknown between susceptible *V. vinifera* and resistant *V. pseudore**ticulata*.

Here, we report two differential expressions of *Why1* genes identified from *V. vinifera* and *V. pseudoreticulata*. Our work shows that both Why1 CDSs have a high identity (99.38%), but with different length of promoter sequences and a lower identity (95.16%). It is the *VpWhy1* promoter (*pVp*) instead of the *VvWhy1* promoter (*pVv*) that drives the *Why1* gene enhanced disease resistance and increases the expression of pathogenesis-related genes (*PR1* and *PR10*) in *Nicotiana benthamiana*. This work adds to the knowledge of the roles of noncoding region regulation in the *Why1* gene and provides a novel insight into disease resistance mechanism.

## 2. Results

### 2.1. Isolation and Characterization of Why1 Genes and Their Promoters

Two *Why1* genes and their 5′-upstream promoter sequences were amplified. *VvWhy1* (GenBank: MN395403) and its promoter sequence (GenBank: MN397251) were isolated from *V. vinifera* “Cabernet Sauvignon” and *VpWhy1* (GenBank: MN395402) and its promoter sequence (GenBank: MN397250) were isolated from *V. pseudoreticulata* “HD1”. Both *Why1* genes had a total length of 862 bp with 807 bp length ORF, encoding 268 amino acid residues (Figure 1a). The *VvWhy1* and *VpWhy1* promoter sequences had 1127 bp and 1136 bp. Interestingly, the *VvWhy1* promoter sequence showed nine bp missing compared with the *VpWhy1* promoter sequence (Appendix A sequence). The sequence alignment showed that the identity between the two *Why1* genes was 99.38%, and their CDS only encoded three different amino acid residues. However, two *Why1* promoters showed a lower identity of 95.16%. In consequence, the sequence alignments showed the *Why1* CDS sequences were more conserved than the *Why1* promoter sequences.

The gene structure analysis showed that both *Why1* sequences had seven exons and six introns (Figure 1b), they had a C-terminal Whirly conserved domain and N-terminal chloroplast transit peptide (Figure 1c), indicating a potential chloroplast subcellular localization. Phylogenetic analyses were conducted to estimate the evolution relationship and an unrooted phylogenetic tree was constructed from an aligned dataset of 27 homologous Why protein sequences (Figure 2). These proteins were grouped into two classes, Why1 and Why2, respectively. Both Why1 and Why2 could be divided into two subgroups, which were derived from woody and herb plants. VvWhy1 from *V. vinifera* and VpWhy1 from *V. pse**udoreticulata* were clustered together with other Why1 of woody plants.

### 2.2. Why1 Expression under P. viticola Induction in V. vinifera and V. pseudoreticulata

To investigate the *Why1* expression model under pathogen infection, the leaves of *V. vinifera* and *V. pseudoreticulata* were inoculated with *P. viticola*. As shown in Figure 3a, the leaf symptoms were significantly different between *V. vinifera* and *V. pseudoreticulata*. *P. viticola* was observed obviously in *V. vinifera* leaves at 4 days postinoculation (dpi), and the leaf symptoms of *P. viticola* aggravated with the extension of the treatment time. However, only a few disease spots appeared in *V. pseudoreticulata* leaves without *P. viticola* growth. The results indicated that *V. vinifera* “Cabernet Sauvignon” was downy-mildew-susceptible, whereas *V. pseudoreticulata* “HD1” was resistant. A transcriptional expression was further performed by using qRT-PCR and the results showed that *VvWhy1* remained at a low expression level (Figure 3b). On the contrary, *VpWhy1* was significantly upregulated under *P. viticola* infection and reached a maximum at 48 h postinoculation (hpi), then rapidly decreased when it was 72 and 96 hpi. A transcriptional expression analysis revealed that *VpWhy1’*s expression was induced by *P. viticola*, while *VvWhy1* remained insensitive to *P. viticola*. *VpWhy1* may play an important role in *P. viticola* resistance.

### 2.3. Subcellular Localization Analysis of Why1

A conserved domain analysis showed the Why1 proteins had an N-terminal chloroplast transit peptide. To verify the subcellular localization of Why1, the plant expression vectors of pBI121-VvWhy1-GFP and pBI121-VpWhy1-GFP were constructed. *Agrobacterium tumefaciens* strain GV3101 harboring a recombinant plasmid was injected into *N. benthamiana* leaves to transiently express the Why1-GFP fusion protein. Results showed that the GFP fluorescence signal of VvWhy1 and VpWhy1 infusion proteins were overlapped with chloroplast autofluorescence signal (Figure 4). Therefore, a subcellular localization indicated that both VvWhy1 and VpWhy1 proteins were chloroplast localization, and may play important roles in chloroplast.

### 2.4. Why1 Promoter Activity Analysis

A previous sequence analysis of two *Why1* genes showed a 99.38% identity and the CDS only encoded three different amino acid residues. The promoter identity (95.16%) was far lower than that in CDS. Moreover, a transcriptional expression analysis showed a different expression pattern of *VvWhy1* and *VpWhy1*. We supposed that the *VvWhy1* promoter (*pVv*) and *VpWhy1* promoter (*pVp*) may play important roles in response to *P. viticola*. To determine the activity of the two promoters, pBI121-*pVv::GUS* and pBI121-*pVp::GUS* were generated and *p0::GUS* and *p35S::GUS* were negative and positive controls (Figure 5a). A GUS histochemical stain showed that the negative control leaves had no staining, and the positive control leaves were stained with deep blue, both controls were not induced by *P. capsici*. The *pVv::GUS* treatment leaves had no staining under the mock treatment and a slight blue stain appeared when it was under *P. capsici* stress. For the *pVp::GUS* treatment, it showed a high GUS activity, as shown by the deep blue staining (Figure 5b). The relative quantitative GUS activity of the *pVv* samples showed no significant difference between the mock and *P. capsici* treatments, but it was significantly induced in the *pVp* treatments (Figure 5c). This study indicated that *pVp* had a high promoter activity and was involved in the response to *P. capsici*.

### 2.5. P. capsici Resistance of Why1 Genes Driven by Native and Exotic Promoters

A transcriptional expression and promoter activity showed that *VpWhy1* and its promoter were strongly induced in response to *P. viticola* and *P. capsici*. To investigate whether the *Why1* and its promoter had *P. capsici* resistance, we generated four Why1 vectors driven by native and exotic promoters (Figure 6a) for the transient expression in *N. benthamiana*. A *P. capsici* resistance assay after trypan blue staining in *N. benthamiana* leaves shown in Figure 6b (left side for *p35* controls, right side for experimental treatments). The leaves exhibited obvious lesion symptoms in the control (left side) and experimental (right side) treatments. For *VvWhy1* and *VpWhy1* driven by *pVp*, the lesion areas were significantly smaller than that in the controls. A further quantitative examination showed that the relative lesion areas in *pVv::VvWhy1* and *pVv::VpWhy1* were smaller than that in *pVp::VvWhy1* and *pVp::VpWhy1*. The result of the lesion areas calculation was consistent with the findings when observing the disease symptoms, which implied that *pVp* significantly enhanced the resistance to *P. capsici*.

### 2.6. Why1 Heterologous Expression and Pathogenesis-Related Genes in Response to P. capsici

The recombinant vectors were described as 2.5 and the transient expression was conducted on both leaf sides. *N. benthamiana* leaf lesion symptoms were observed by using a UV light under *P. capsici* infection at 36 and 54 hpi (Figure 7a). All treatments of *N. benthamiana* displayed varying degrees of disease symptoms. The *pVp* treatments of transient transgenic leaves had a smaller lesion area than the control (*p35S*) and *pVv* treatments at 36 hpi. The lesion area expanded in all leaf samples with different treatments at 54 hpi. Then, a disease resistance evaluation was conducted by calculating the lesion length in *N. benthamiana*. The lesion length of *p35S* and the two *pVv* treatments were between 2.14 and 2.27 cm, while the two *pVp* treatments were between 1.68 and 1.75 cm, respectively. The lesion length of the two *pVv* treatments were significantly longer than that in the two *pVp* treatments, these results were consistent with those of the above trypan blue staining assay, indicating that *pVp* enhanced *P. capsici* resistance.

To further explore the molecular mechanism of *pVp* in the resistance to *P. capsici*, a transcriptional expression of *Why1* and *PR* genes was performed in transient transgenic *N. benthamiana*. *Why1* and *PR* genes could be divided into two groups; *Why1*, *PR1* and *PR10* were strongly induced in *pVp* transient transgenic *N. benthamiana*, whereas *PR2*, *PR4* and *PR5* remained at a relative low expression level. The expression of *PR*2, *PR4* and *PR5* showed an increasing trend with time extension and were not significantly induced by different transgenic treatments under *P. capsici* infection. *Why1*, *PR1* and *PR10* were significantly induced in *pVp* transient transgenic treatments, while the expression remained at a lower level in the *pVv* treatments. *VvWhy1* and *VpWhy1* were upregulated, and reached a maximum expression level exceeding 14- and 15-fold change at 24 and 36 hpi in the *pVp::VvWhy1* and *pV**p::VpWhy*1 transgenic treatments, respectively. The *PR1* gene was upregulated and reached a maximum at 24 dpi, then declined at 36 dpi. The expression of *PR10* increased dramatically at 6 hpi in both *pVp* treatments. For the *pVp::VvWhy1* treatment, *PR10* continued to increase and reached a maximum expression level. However, the *PR10* expression stayed at a stable level, and even decreased at 36 hpi. As a result, *pVp* enhanced the expression of *Why1*, *PR1* and *PR10*; these findings were in accordance with the disease symptoms observation in the *P. capsici* resistant assay. In conclusion, these experiments suggested that *pVp* promoted the *Why1* transcription. Furthermore, *Why1* regulated the expression of *PR1* and *PR10*, which finally enhanced the resistance to *P**. capsici*.

## 3. Discussion

### 3.1. Dual Localization of Why1

Whirly (Why) proteins appeared to have multiple functions in plant growth and development, biotic and abiotic stress, which may attribute to their complicated subcellular localization. Desveaux [19] provided clear evidence for an insight into a novel single-stranded DNA-binding transcription activator PBF-2 (StWhy1), which was identified from *Solanum tuberosum*. The plant did possess at least two Why members, and a previous study indicated that in the cases of two members, the Why proteins were putatively chloroplast- (Why1) and mitochondrial-localized (Why2), respectively [21]. The occurrence of the third Why was revealed to be nucleus-localized and may be restricted to *Arabidopsis* [33].

However, recent studies have revealed that Why1 translocated from chloroplast to nucleus [34]. Proteins with dual subcellular localization mediate diverse intercellular signaling processes and various functions [35,36,37]. As a consequence, Why1 is dual-localized to the chloroplast and nucleus [38,39]. The translocation function may result from stress-associated redox changes in the photosynthetic apparatus [40]. Moreover, the phosphorylation and oxidization of the Why1 protein also lead to different subcellular localization in the nucleus or plastid [38,41]. Why1 is an excellent candidate for communication between chloroplasts and nucleus due to its dual localization in chloroplasts and nucleus [42]. Why1 has even been shown to be relocated from one compartment to another upon environmental or developmental clues [43]. In our study, VvWhy1 and VpWhy1 showed chloroplast-localization in *N. benthamiana*, and no fluorescence signal was detected in the nucleus. Similarly, it raised the question of whether it is dual-targeted in vivo since a Why1-GFP protein in potato mesophyll protoplasts localized to the chloroplasts but not to the nucleus in an earlier study. According to the translocation and relocalization functions, Why1 is a sequestered nuclear transcription factor that can be released from plastids under certain conditions [43]. We hypothesized that stress treatments may lead to the dual localization of VvWhy1 and VpWhy1 proteins.

### 3.2. Promoter Differentiation between VvWhy1 and VpWhy1

*Vitis* plants including Eurasian, East Asian and North American species share a close genetic relationship and conservative genomic sequences. The genome sequencing completion of the Eurasian grapevine *V. vinifera* “Pinot Noir” marks a new stage of grapevine research [44]. The “Pinot Noir” genome is regarded as a reference for other grapevine species in genome assembly, gene annotation, gene cloning and function verification based on their close genetic backgrounds. However, different grapevines show a varied degree of disease resistance; *V. vinifera* “Cabernet Sauvignon” is downy-mildew-susceptible, while *V. pseudoreticulata* “HD1” is resistant [45].

In this study, both grapevines showed a different response to *P. viticola*. However, *Why*1 CDS from *V. vinifera* and *V. pseudoreticulata* showed a high identity of 99.38% with five different bases, which encoded three different amino acids. The VvWhy1 and VpWhy1 proteins had similar physical and chemical properties. In contrast, the upstream promoter regions of both grapevines possessed different sequence lengths: 1127 bp and 1136 bp promoter sequences were obtained from *V. vinifera* and *V. pseudoreticulata*, respectively. The *VvWhy1* promoter (*pVv*) sequence was nine bp shorter than that in the *VpWhy1* promoter (*pVp*). Moreover, the promoter sequence identity was 95.16%, which was far lower than that in CDSs. Therefore, *VvWhy1* and *VpWhy1* had conserved CDS but with promoter sequence differentiation. The different response to *P. viticola* between *VvWhy1* and *VpWhy1* may result from the promoter differentiation, and further lead to the disease resistance differentiation in different grapevines. In general, promoter sequence alterations provided an effective manner to regulate gene expression and function. Interestingly, the homologous genes *VvSTS* and *VpSTS* from *V. vinifera* “Carigane” and *V. pseudoreticulata* share >99% identity on the amino acid level, but with significant difference in promoter regions; *VpSTS* conferred powdery mildew resistance with an elevated responsiveness of the promoter [46]. In *V. labrusca* “Concord”, the absence of a 426 bp and/or a 42 bp sequence in the acyltransferase gene (AMAT) promoter highly associated with high levels of AMAT expression, and further regulated the methyl anthranilate (MA) accumulation, which caused a special “foxy” aroma [47]. We claim that promoter differentiation between *VvWhy1* and *VpWhy1* may lead to disease resistance differentiation in *V. vinifera* and *V. pseudoreticulata.*

### 3.3. Why1 Driven by VpWhy1 Promoter (pVp) Enhanced Disease Resistance via Regulating the Expression of PR Genes

The *Why1* gene function has been elucidated in multiple biological functions, but the transcriptional regulation of its noncoding region has rarely been reported. Based on the high conservation of *Why1* CDS and promoter differentiation between *VvWhy1* and *VpWhy1*, the differential expression of *VvWhy1* and *VpWhy1* in response to pathogen may result from their promoter sequence. The activity of the *VpWhy1* promoter was significantly higher than that of the *VvWhy1* promoter and was induced by *P. viticola* and *P. capsici*. *VvWhy1* and *VpWhy1* driven by the *VpWhy1* promoter (*pVp*) increased *P. capsici* resistance in *N. benthamiana* leaves. StWhy1 participated in interactions with the elicitor response element (ERE) of *PR-10a*, its binding to the ERE correlating with the expression of *PR-10a* [19]. The transcription activator *StWhy1* encoding a protein of 24 kD, critical for the interaction of PBF-2 with single-stranded DNA or RNA is a motif consisting of KGKAAL. Mutations in this domain have occurred in the absence of DNA-binding activity [48]. Why1 also plays a new role in salicylic acid (SA) biosynthesis via the coordination of isochorismate synthase1 (ICS1), phenylalnine ammonialyase (PAL1) and S-adenosyl-L-Met-dependent methyltransferase1 (BSMT1) [39]. SA plays important roles in disease resistance and pathogenesis-related response.

Pathogenesis-related genes (*PR*) are among the best characterized genes induced by pathogens [19]; it is one of the key components of plant innate immune system especially systemic acquired resistance (SAR) [49]. Rossarolla [13] indicated the stronger and earlier activation of the defense pathway in the genotypes containing pyramided *Rpv* was evidenced in the expression of the evaluated *PR* genes, among which *PR1* and *PR10,* involved in the defense reaction, were mainly associated with the SA signaling pathway and JA signaling pathway, respectively [50,51]. The overexpression of *VpPR10.1* from *V. pseudoreticulata* enhanced downy mildew resistance in *V. vinifera* [52]. *PR* genes, including *PR-1*, *PR-3*, *PR-10,* contributed to powdery mildew resistance in wheat [53]. *Arabidopsis Why1* was reported to be involved in disease defense signaling and required for pathogen-induced *PR1* expression [33]. To further investigate whether the *PR* genes were regulated by *VvWhy1* and *VpWhy1* under *P. capsici* infection, a transcriptional expression of *PR1*, *PR2*, *PR4*, *PR5* and *PR10* was conducted in *N. benthamiana* transiently expressing *VvWhy1* and *VpWhy1* driven by *pVv* and *pVp*. The results showed that *pVp* promoted *VvWhy1* and *VpWhy1* correlated with the expression of *PR1* and *PR10*. As a result, *Why1* driven by the *VpWhy1* promoter (*pVp*) enhanced disease resistance via regulating the expression of *PR1* and *PR10* genes.

## 4. Materials and Methods

### 4.1. Plant Materials, Pathogen and Treatments

One-year-old *V. vinifera* “Cabernet Sauvignon” and *V. pseudoreticulata* “HD1” from Shangzhuang experimental station of China Agricultural University, were grown in pots at 25 ± 2 °C under a 16/8 h light/dark photoperiod in a greenhouse. For real-time PCR, grapevine leaves were inoculated with *P. viticola* sporangia suspension liquid (10^5^ sporangia mL^−1^). Five grapevine leaf samples were collected at 0, 6, 12, 24, 48, 72, and 96 hpi. One-month-old *N. benthamiana* seedlings were used for transient gene expression and disease resistance evaluation, the third to fifth unfolded leaves from the shoot apex were injected with *A. tumefaciens* transient suspension liquid harboring a recombinant plastid. After 48 h, five *N. benthamiana* leaves were excised for each *P. capsici* inoculation treatment. Three biological replicates were performed for *P. viticola* and *P. capsici* inoculation.

### 4.2. Gene Cloning and Sequence Analysis

*V. vinifera* “Cabernet Sauvignon” and *V. pseudoreticulata* “HD1” genomic DNA and total RNA were isolated according to the CTAB method with minor modifications. Reverse transcription was performed using RevertAid TM First Strand cDNA Synthesis Kit (Invitrogen, Waltham, MA, USA). Genomic DNA and cDNA were used for promoter and gene cloning, respectively. Primers were designed according to predicted *Why1* gene sequences and their upstream regulatory sequences downloaded from Phytozome *V. vinifera* “Pinot Noir” reference genome; primer sequences are listed in the Appendix A. *Why1* genes and promoters were cloned by using PCR amplification, electrophoretic band purification, cloning vector ligation, *Escherichia coli* transformation and finally sequencing validation with four *E. coli* clones. The gene structure was analyzed by online software Gene Structure Display Server (GSDS) [54]. The conserved domain was predicted using online NCBI-CDD [55]. The phylogenetic tree was constructed using MEGA 5.0 with the neighbor-joining (NJ) method. The promoter sequences were analyzed using PlantCARE [56] to search for *cis*-acting elements.

### 4.3. Quantitative Real-Time PCR Analysis of Why1 under P. viticola Induction

Total RNA was extracted from grapevine using the CTAB method with minor modification. The cDNA synthesis was conducted according to PrimeScript™ RT reagent Kit (Takara, Kusatsu, Japan). Three biological cDNA replicates were mixed for next amplification. Specific primers listed in Appendix A were designed for real-time PCR. The amplification was performed on the QIAGEN Rotor-Gene Q system (QIAGEN, Hilden, Germany) using the SuperReal PreMix Plus (SYBR Green) kit (TIANGEN, Beijing, China). Constitutively expressed elongation factor1-α (EF1-α) was used as reference gene to calculate relative expression levels with the 2^−∆∆Ct^ method. All reactions were performed with three technical replicates.

### 4.4. Subcellular Localization Analysis

Plant expression vectors pBI121-*VpWhy1*-*GFP* and pBI121-*VvWhy1*-*GFP* driven by the *CaMV 35S* promoter were constructed for the subcellular localization analysis. Primers with *Bam*HI and *Kpn*I (2 bp insertion was required to avoid frameshift mutation) restriction sites flanking the *VvWhy1* and *VpWhy1* sequences were designed (Appendix A). Ten positive *E. coli* clones harboring recombinant plasmids were validated by PCR, among which four clones were further sequenced. The recombinant plasmids were introduced into *Agrobacterium tumefaciens* (GV3101) via electroporation. The GV3101 was harvested when the OD_600_ reached ~1.0 grown with rifampicin (50 mg/L) and kanamycin (60 mg/L) in LB liquid medium, and then resuspended with MMA solution supplemented with 10 mM MgCl_2_, 10 mM MES, pH 5.7 and 100 μM acetosyringone, to a final OD_600_ of 0.6. The Why1-GFP fusion proteins were transiently expression in *N. benthamiana* using the *Agrobacterium*-mediated leaf injection method as previously described [57]. Fluorescence observation was conducted using an FV1000 confocal laser-scanning microscope (Olympus, Tokyo, Japan).

### 4.5. Promoter Activity Analysis

To estimate the *Why1* promoter activity, primers with *Hin*dIII and *Sma*I restriction sites flanking the *VvWhy1* promoter (*pVv*) and *VpWhy1* promoter (*pVp*) were designed to construct pBI121-*pVv*::*GUS* and pBI121-*pVp*::*GUS*, while pBI121-*p0*::*GUS* and pBI121-*35S*::*GUS* were negative and positive controls, respectively. The method of GUS transiently expression in *N. benthamiana* was described as above. The pathogen stress was conducted with *P. capsici* zoospore suspensions according to [58]. The GUS histochemical staining and quantitative detection were carried out to qualitatively and quantitatively assess the promoter activity, respectively. A fluorescence spectrophotometer was used for the GUS quantitative determination as described in [59]. The total protein concentration was measured with the Bradford method [60]. Three biological replicates were performed for the GUS histochemical staining and quantitative detection.

### 4.6. P. capsici Resistance Assay Transiently Expressing Why1 Driven by Native and Exotic Promoter

For the *P. capsici* resistance assay, based on previous constructs of pBI121-*VvWhy1*-*GFP* and pBI121-*VpWhy1*-*GFP* in subcellular localization, primers with *Sbf*I and *Bam*HI flanking *pVv* and *pVp* were designed to construct 4 pBI121 plant expression vectors containing *pVv*::*VvWhy1*, *pVv*::*VpWhy1*, *pVp*::*VvWhy1* and *pVp*::*VpWhy1*, and the *CaMV 35S* promoter was used as control. Transient expression was conducted as mentioned above. A *P. capsici* resistance assay was performed with the disk inoculation method according to a previous protocol [17]. To estimate the resistance to *P. capsici* of *Why1*, trypan blue staining was carried out and the disease lesion area was calculated. The relative lesion area is the ratio of the experimental lesion areas to those on the control. Three biological replicates were used, each of which contained five leaf samples.

### 4.7. Why1 Heterologous Expression and Pathogenesis-Related Genes in Response to P. capsici

To further illustrate the molecular mechanism of disease resistance, the expression of *Why1* and pathogenesis-relative genes (*PR*) was performed. The plant expression vectors and transient transformation procedure were described as the above transient expression in the *P. capsici* resistance assay, the difference being that *N. benthamiana* was injected with the same *Agrobacterium* suspensions on both leaf sides. *P. capsici* infection was conducted after 48 h of transient expression. Five detached leaves were collected when it was 0, 6, 12, 24 and 36 hpi for the transcriptional expression analysis, using three biological replicates. The qRT-PCR of *Why1* associated with *PR1*, *PR2*, *PR4*, *PR5* and *PR10* in transgenic *N. benthamiana* was performed; the primer sequences are listed in Appendix A. Lesion symptoms were observed by using a UV light, and the lesion lengths were calculated to estimate the disease resistance with five leaf samples for each treatment and three biological replicates.

### 4.8. Statistical Analysis

The quantitative results for the gene expression analysis, GUS activity, lesion area and lesion length are presented as the means ± standard deviations (SD). Tukey’s multiple comparisons test was conducted by using GraphPad Prism 8.0. Different letters on the columns indicate statistically significant differences at the 0.05 level. Figures were prepared using GraphPad Prism 8.0.

## 5. Conclusions

The single-stranded DNA- and RNA-binding protein genes *VvWhy1* and *VpWhy1* with promoter sequences were isolated from *V. vinifera* “Cabernet Sauvignon” and *V. pseudoreticulata* “HD1”. *VpWhy1* was strongly induced by *P. viticola*, while *VvWhy1* showed a low expression level. *VvWhy1* and *VpWhy1* shared 99.38% of identity encoded in only three different amino acid residues. However, the identity of the promoter sequences was far lower than that between *Why1* CDSs. Therefore, we proposed that the differentiation of *Why1* promoters played important roles in pathogen response. Further, the GUS activity indicated the *VpWhy1* promoter (*pVp*) had a high activity involved in response to *P. capsici* infection. *N. benthamiana* transiently expressing *pVp*::*Why1* (*pVp*::*VvWhy1* and *pVp*::*VpWhy1*) enhanced the *P. capsici* resistance. Moreover, *Why1*, *PR1* and *PR10* were upregulated under *P. capsici* infection in *pVp* transient transgenic *N. benthamiana* leaves. *PR* genes were reported to be regulated by *Why1*; in consequence, *PR1* and *PR10* were promoted by *pVp*::*Why1*. In conclusion, this study provides new insight into the novel disease resistance mechanism of *VpWhy1* with a strong pathogen-induced promoter.

## Figures and Tables

**Figure 1 ijms-23-08052-f001:**
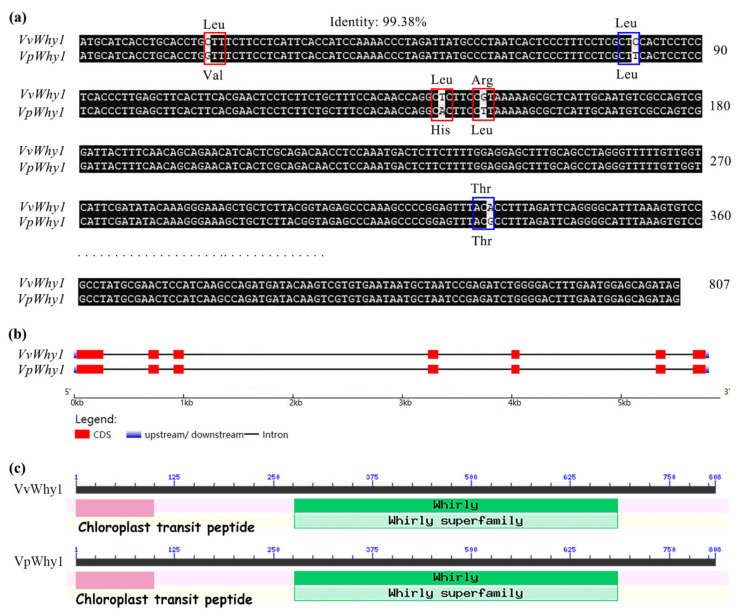
Sequence alignment, gene structure and conserved domain of *VvWhy1* and *VpWhy1*. (**a**) Sequence alignment of *VvWhy1* and *VpWhy1* shows an identity of 99.38% with five different nucleotides encoding three different amino acid residues. The red frame shows the encoding of different amino acids, and the blue frame shows the encoding of the same amino acids. (**b**) Gene structure of *VvWhy1* and *VpWhy1*, which consisted of seven exons and six introns. (**c**) Conserved domain of VvWhy1 and VpWhy1 reveals C-terminal Whirly conserved domain and N-terminal chloroplast transit peptide.

**Figure 2 ijms-23-08052-f002:**
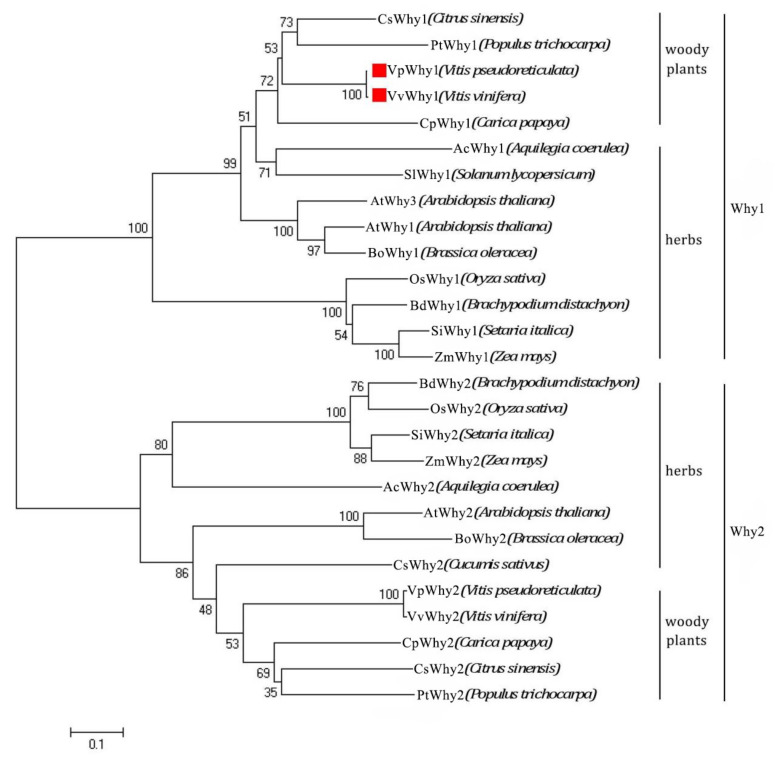
Phylogenetic tree construction of Why family. A dataset of 27 homologous Why protein sequences were constructed using MEGA 5.0 with neighbor-joining; the numbers at the nodes represent the bootstrap values based on 1000 replications. VvWhy1 from *V. vinifera* and VpWhy1 from *V. pseudoreticulata* were clustered together with other Why1 proteins from woody plants.

**Figure 3 ijms-23-08052-f003:**
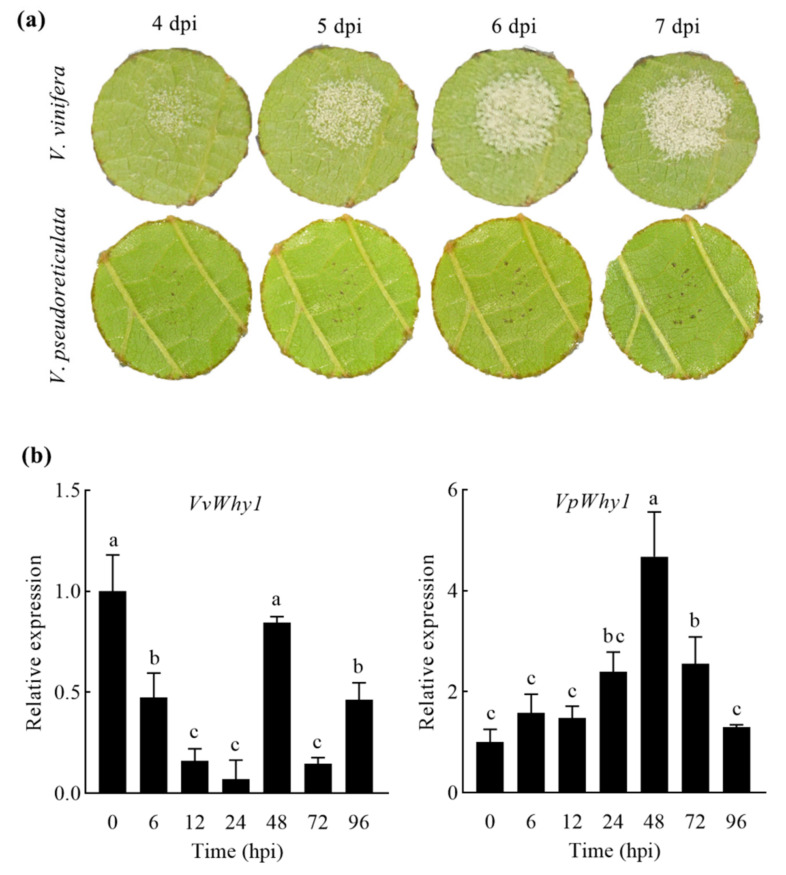
Disease symptoms and relative expression of *VvWhy1* and *VpWhy1* in *V. vinifera* “Cabernet Sauvignon” and *V. pseudoreticulata* “HD1” under *P. viticola* infection. (**a**) Disease symptoms show *V. vinifera* and *V. pseudoreticulata* are downy-mildew-susceptible and resistant, respectively. (**b**) Relative expression of *VvWhy1* and *VpWhy1*. *VpWhy1* is strongly induced in response to *P. viticola*, while *VvWhy1* is insensitive to *P. viticola*. hpi indicates hours postinoculation. Tukey’s multiple comparisons test was conducted using GraphPad Prism 8.0. Vertical bars represent standard deviations; different letters indicate significant differences at the 0.05 level.

**Figure 4 ijms-23-08052-f004:**
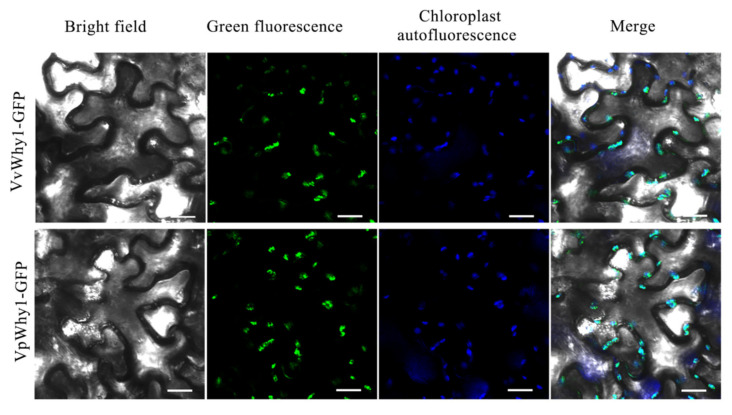
Subcellular localization of VvWhy1 and VpWhy1. *N. benthamiana* leaves were used to express Why1-GFP fusion protein. Fluorescence observation was conducted using an FV1000 confocal laser-scanning microscope (Olympus, Tokyo, Japan). The green signal indicates GFP fluorescence, the blue signal indicates chloroplast autofluorescence and the cyan signal indicates merge result. Both Why1 proteins were chloroplast-localized. Scale bar = 20 μm.

**Figure 5 ijms-23-08052-f005:**
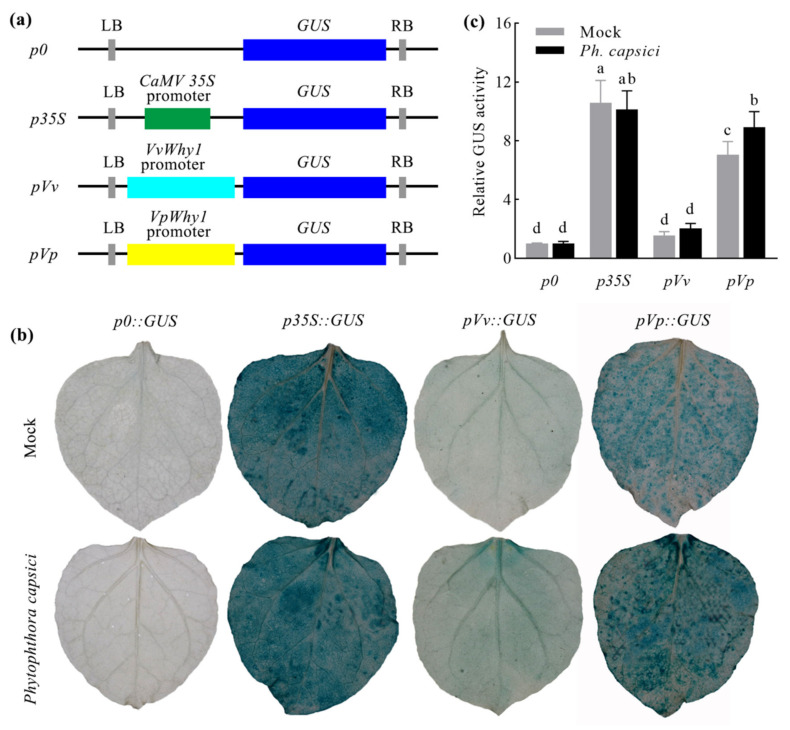
Construction of plant expression vectors and analysis of promoter activity. (**a**) Schematic representation of *Why1* promoter-GUS constructs. *p0* and *p35S* refer to negative and positive controls, respectively. (**b**) Histochemical staining analysis of GUS activity in transiently transformed *N. benthamiana* leaves. *P. capsici* treatments were prepared with zoospores (100 zoospores/μL), while mock treatments were sprayed with sterile water. (**c**) Relative quantitative GUS activity of fluorometric analysis. Tukey’s multiple comparisons test was conducted by using GraphPad Prism 8.0. Vertical bars represent standard deviations; different letters indicate significant differences at the 0.05 level.

**Figure 6 ijms-23-08052-f006:**
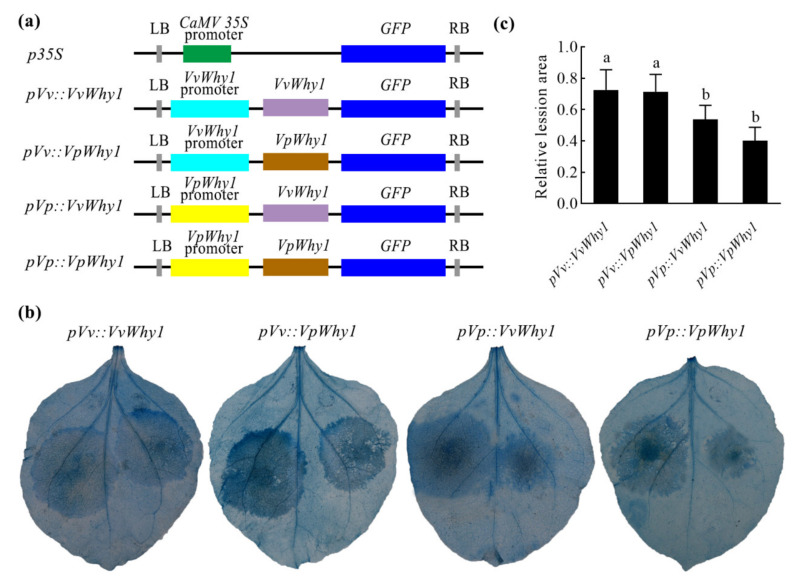
Construction of plant expression vectors and trypan blue staining assay in transient expression *N. benthamiana* leaves under *P. viticola*. (**a**) Schematic representation of *Why1* driven by native and exotic promoters. (**b**) Trypan blue staining assay in transient expression *N. benthamiana* leaves at 54 hpi. The left side of leaves represent control treatments (*p35S*), those on the right side represent transient expression treatments. (**c**) Relative lesion area of transient expression *N. benthamiana* leaves. The relative lesion area was the ratio of the lesion area on transient expression leaves to those on the controls. Vertical bars represent standard deviations; different letters indicate significant differences at the 0.05 level.

**Figure 7 ijms-23-08052-f007:**
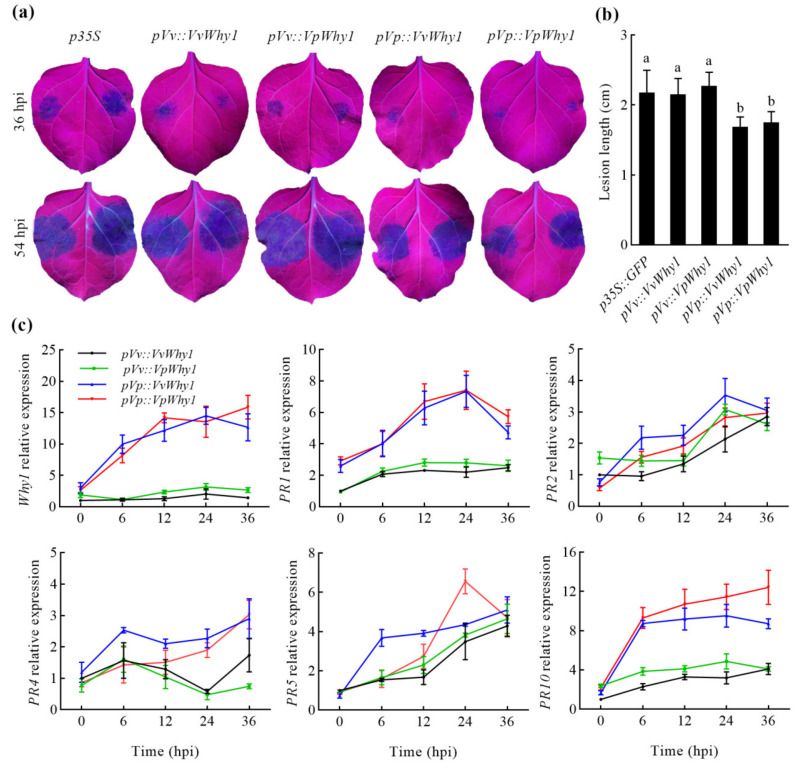
Lesion symptoms of transient transgenic *N. benthamiana* and transcriptional expression of *Why*1 and *PR* genes in response to *P. capsici*. (**a**) Lesion symptoms of transient transgenic *N. benthamiana* at 36 and 54 hpi. (**b**) Lesion length of transient transgenic *N. benthamiana* leaves (different letters indicate significant difference at 0.05 level). (**c**) Transcriptional expression of *Why*1 and *PR* genes in response to *P. capsici*. *Why1*, *PR1* and *PR10* are significantly induced in *pVp* transient transgenic treatments, while the expression remains at a lower level in *pVv* treatments. Vertical bars represent standard deviations; different letters indicate significant differences at the 0.05 level. hpi indicates hours postinoculation.

## Data Availability

Not applicable.

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
