# Peer review of "The Single-Stranded DNA-Binding Gene *Whirly* (*Why1*) with a Strong Pathogen-Induced Promoter from *Vitis pseudoreticulata* Enhances Resistance to *Phytophthora capsici"

_ijms, 2022, doi:10.3390/ijms23148052_

Round 1

Reviewer 1 Report

The manuscript ijms-1785706 entitled “The Single-Stranded DNA-Binding Gene Whirly (Why1) with a Strong Pathogen Induced Promoter from Vitis pseudoreticulata Enhances Resistance to Phytophthora capsici” investigated the expression of gene VvWhy1 and VpWhy1 induced by Plasmopara viticola in both resistant and susceptible Vitis species and their subcellular localization. Furthermore, the activity of promoters of VvWhy1 (pVv) and VpWhy1 (pVp) in response to Phytophthora capsici was tested in Nicotiana benthamiana. Promoter activity was found for pVp, suggesting its involvement in Ph. capsici resistance.

The manuscript is well organized, but relevant information in methods is missing. In particular, the number of biological repetitions collected for each leaf sampling are not indicated in the manuscript. Statistical analyses of gene expression are referred to biological or technical repetitions? The statistical tests used for the analysis reported in fig. 3b - 5c - 6c - 7b are missing, a specific paragraph on data analysis should be added in Material and Methods. Any statistical test was performed on data showed in fig. 7c? In my opinion, these information should be provided before to consider the manuscript for publication in high quality journal as IJMS.

Other minor issues are related to missing references (lines 31-34 and lines 344-346).

In the introduction, I would refer to the three main Vitis clusters as clades rather than populations, as reported in literature (Wan et al., 2013; Liu et al., 2016).

Finally, I suggest authors to add final conclusions for this study in a separated paragraph.

Author Response

Dear Editor and reviewer,

Thank you for your letter and for the reviewers’ comments concerning our manuscript entitled “The Single-Stranded DNA-Binding Gene Whirly (Why1) with a Strong Pathogen Induced Promoter from Vitis pseudoreticulata Enhances Resistance to Phytophthora capsici” (Manuscript ID: ijms-1785706). Those comments are all valuable and very helpful for revising and improving our paper, as well as the important guiding significance to our researches. We have considered each of the comments carefully and the itemized responses to your comments are outlined below. We used the "Track Changes" marked the changes in the revised manuscript for easily visible. We believe that the manuscript has been significantly improved by the suggested changes.

Response to the reviewers’ comments:

  1. The manuscript is well organized, but relevant information in methods is missing. In particular, the number of biological repetitions collected for each leaf sampling are not indicated in the manuscript.

Response: We have added the leaf sampling and biological repetition information in the methods.

  1. Statistical analyses of gene expression are referred to biological or technical repetitions?

The statistical tests used for the analysis reported in fig. 3b - 5c - 6c - 7b are missing, a specific paragraph on data analysis should be added in Material and Methods.

Response: A specific paragraph of “Statistical Analysis” was added in the end of Material and Methods. For gene expression, cDNA extraction was performed with three biological replicates, and were mixed for next amplification. Amplification reactions were performed with three technical replicates. Relevant statistical information was added in fig. 3b - 5c - 6c - 7b.

  1. Any statistical test was performed on data showed in fig. 7c?

Response: Relevant statistical information was added in fig. 7c .Vertical bars represent standard deviations, Tukey’s multiple comparisons test was conducted using GraphPad Prism 8.0. Different letters indicate significant differences at the 0.05 level.  

  1. Other minor issues are related to missing references (lines 31-34 and lines 344-346).

Response: Four related references were added to lines 31-34 and lines 344-346.

  1. In the introduction, I would refer to the three main Vitis clusters as clades rather than populations, as reported in literature (Wan et al., 2013; Liu et al., 2016).

Response: The mentioned sentence was changed to “Vitis plants are assigned to species originated in three regions, Eurasia, East Asian and North American”, and related reference was added.

  1. Finally, I suggest authors to add final conclusions for this study in a separated paragraph.

Response: We have added a separated paragraph of “Conclusions”.

Reviewer 2 Report

My comments and suggestions are included in the manuscript file.

Author Response

Dear Editor and reviewer,

Thank you for your letter and for the reviewers’ comments concerning our manuscript entitled “The Single-Stranded DNA-Binding Gene Whirly (Why1) with a Strong Pathogen Induced Promoter from Vitis pseudoreticulata Enhances Resistance to Phytophthora capsici” (Manuscript ID: ijms-1785706). Those comments are all valuable and very helpful for revising and improving our paper, as well as the important guiding significance to our researches. We have considered each of the comments carefully and the itemized responses to your comments are outlined below. We used the "Track Changes" marked the changes in the revised manuscript for easily visible. We believe that the manuscript has been significantly improved by the suggested changes.

Response to the reviewers’ comments:

Reviewer 2:

  1. first mention in full, numbers smaller than 10 should be in full across lthe text,except in from of units, The authors should follow the botanical taxonomy guidelines, across the text.

Response: We have updated the full name which was first mentioned in the manuscript. Numbers smaller than 10 were also updated. Pl. viticola and Ph. capsici were revised as P. viticola and P. capsici, respectively.

  1. In the introduction, it is lacking mention to the 31 Rpv genes that the authors should refer to as well. Examples of articles that can be a source of the lacking information: ÿttps://doi.org/10.3389/fpls.2021.693887;https://doi.org/10.5073/vitis.2021.60.195-206

Response: Thanks for providing the valuable reference, we added the information of 31 rpv and related references including above mentioned.

  1. While two Why1 promoters showed 95.16% identity needs to be rewritten.

Response:This sentence was rewritten as “But two Why1 promoters showed lower identity with 95.16%”.

  1. Incomplete legend in Figure 2, 3, 4, 5, 6, 7.

Response: We added the relevant information of above mentioned figures, such as the letters, bars, the tested genotypes.

  1. Confuse sentence of “In conclusion, these experiments suggested that pVp promoted the Why1 transcription, Why1 subsequently regulated the expression of PR1 and PR10, which further enhanced the resistance to Phytophthora capsici.”

Response: This sentence was rewritten as “In conclusion, these experiments suggested that pVp promoted the Why1 transcription. Furthermore, Why1 regulated the expression of PR1 and PR10, which finally enhanced the resistance to Phytophthora capsici.”

  1. Genotype or accession name of “V. pseudoreticulata” (line 267)? The authors tested just one genootype for each species, then they cannot claim for the all other varieties of plant populations.

Response: “V. pseudoreticulata” was updated as “V. pseudoreticulata ‘HD’”.

  1. instead As a result, the authors could claim that...

Response: We have revised the manuscript as the reviewer suggested.

  1. The discussion should be enriched with articles on gene expression of Rpv genes; e.g.https://doi.org/10.5073/vitis.2021.60.195-206, among others

Response: The reviewer provided a valuable reference, and we have added the related gene expression information.

  1. Leaf samples were collected at 0, 6, 12, 24, 48, 72, and 96 hpi How many replications? Statistical analysis?

Response: Five grapevine leaf samples were collected at different time with three biological replicates. A specific paragraph of “Statistical Analysis” was added in the end of Material and Methods.

  1. How many times sequences were sequenced?finally sequencing verification.

Quantitative Real-Time PCR Many information is lacking here.

Response: Sequencing validation with four E. coli clones were performed. Quantitative Real-Time PCR Information was added in the revised manuscript.

  1. The recombinant plasmids were validated by PCR and sequencing, how many?

Response: Ten positive E. coli clones harboring recombinant plasmids were validated by PCR, among which four clones were further sequenced.

  1. Lesion symptoms were observed by using UV-light, lesion length were calculated to estimate the disease resistance. Number of samples and statistical analysis?

Response: Five leaf samples for each treatment, three biological replicats. A specific paragraph of “Statistical Analysis” was added in the end of Material and Methods.

Round 2

Reviewer 1 Report

The missing information has been provided, in my opinion the manuscript ijms-1785706 can be accepted for publication in IJMS